# Communication Skills and Transformational Leadership Style of First-Line Nurse Managers in Relation to Job Satisfaction of Nurses and Moderators of This Relationship

**DOI:** 10.3390/healthcare9030346

**Published:** 2021-03-18

**Authors:** Nadežda Jankelová, Zuzana Joniaková

**Affiliations:** Department of Management, Faculty of Business Management, University of Economics in Bratislava, Dolnozemská cesta 1, 852 35 Bratislava, Slovakia; nadezda.jankelova@euba.sk

**Keywords:** job satisfaction of nurses, communication skills, transformational leadership style, management practice, span of control, psychosocial work factors

## Abstract

The job satisfaction of nurses is reflected in almost all organizational outputs of medical facilities. First-line nurse managers (FLNMs), who are directly related to subordinate nurses, have a great influence on this satisfaction. The aim of our paper is to examine the connection between communication skills and the transformation style of FLNMs management with the job satisfaction of nurses and to verify the influence of three moderators on the strength of this relationship. The chosen moderators—the practice of managing FLNMs, the degree of control (span of control) and psychosocial work—follow from theoretical studies. The moderating effect of the variable management practice is also significant from the point of view of Slovak legislation. The sample consisted of 132 FLNMs from five university hospitals in Slovakia. Data collection took place in the form of a questionnaire. All data were processed using the SPSS 24 software package. A series of regression analyzes were used to identify the proposed hypotheses. ANOVA analysis was used to analyze multiple dependencies. We worked at a 5% level of significance. The findings point to the strong direct effects of communication skills and the transformational leadership style of FLNMs on nurses’ job satisfaction. Moderation effects are mild, but significant in the case of management and span of control practices. The lower values of both variables reinforce the positive relationships among the two predictors and the job satisfaction of nurses. The third moderator, psychosocial work factors, also have a significant moderating effect, which is negative, and the higher value of this moderator mitigates both positive direct effects.

## 1. Introduction

Global healthcare is in the midst of major changes caused by the current economic challenges facing the world. We are in an environment of a changing demographic situation, namely, an aging population in the most developed countries and a growing global population. This dynamic environment places high demands on healthcare and requires its effective management [1].

One of the current problems of healthcare and nursing is the high rate of burnout of medical staff and their low job satisfaction. These facts are currently also compounded by the high workload and workload of managing the global COVID-19 pandemic. The sustainability of the healthcare workforce is, thus, likely to become a major issue in the coming years, which healthcare facilities will be forced to address [2].

First-line nurse managers play an important role in solving these problems of nursing. It is often their responsibility to establish meaningful connections with patients, subordinate nurses and all professionals involved. In this way, they build bridges and maintain interprofessional relationships, which are generally lacking in most clinical microsystems. FLNMs are a critical link for interdisciplinary collaboration and shared decision-making [3]. They directly influence the work of the nurses themselves; through their leadership, support and creation of ideal conditions for work and learning, they can significantly strengthen their position and achieve an increase in their job satisfaction and work performance [4]. According to Rouse and Al-Maqbali [5], FLNMs also play an extremely important role in creating a healthy work environment in nursing. They serve as a model, which sets the level and expectations for healthcare organizations. However, according to Lewis and Malecha [6], up to 68% of nurses report a negative experience with their immediate superiors.

In the environment of Slovak nursing, the situation is specific, due to the financial underestimation of both nurses and FLNMs, whose salaries have remained low for a long time. To ensure quality healthcare, it is, therefore, necessary to look for additional motivational tools for these key employees. The FLNMs management role is also at the forefront in this area. As part of a pilot survey conducted in Slovak health facilities among nurses over three years, we found that at the top of the identified problems in their work environment was the level of leadership and communication with direct superiors, i.e., FLNMs.

Conducted scientific studies have confirmed the positive effects of FLNMs leadership skills on nurses’ performance and satisfaction. According to Morsiani et al. [7] and Vesterinen et al. [8], the leadership style of FLNMs affects staff satisfaction and commitment. It is clear that the head nurse’s trust in subordinate employees supports their motivation and commitment to work. Doran et al. [9] stated that especially transformational leadership in the nursing environment has a significant positive impact on employee satisfaction, creating a favorable environment in the workplace which facilitates necessary cooperation, improves teamwork and reduces conflicts. Professional management of FLNMs also reduces nurses’ stress feelings [10,11] and promotes their stabilization [12], induces a safe working climate, which is ultimately associated with reduced reporting of errors and emotional exhaustion of nurses [13].

The competence of FLNMs in communication is also a prerequisite for the quality of health care provided. The role of communication in promoting organizational efficiency is increasingly recognized and emphasized [14]. The absence of effective communication can compromise patient safety and quality of care, so it is the responsibility of FLNMs to ensure effective communication, as well as to develop and maintain communication skills in the clinical setting [15]. Parson and Stonestreet [16] reported that FLNMs communication skills, their ability to listen, to articulate expectations clearly and to provide feedback have a significant impact on nurses’ job satisfaction. Effective supportive communication in the nursing environment is an extremely important tool; according to the findings of Rouse and Al-Maqbali [5], it promotes the dignity and respect of nurses and is critical for building trust [17].

Several presented studies [18] confirmed the connection between the leadership skills and communication skills of FLNMs and the job satisfaction of nurses. In turn, this is reflected in the patient’s perception of the level of quality of healthcare [19,20]. The main predictors of nurses’ job satisfaction include senior leadership style [21], degree of responsibility [22] and decision-making autonomy [23], workload and lack of recognition [24,25] and open communication by FLNMs [16]. Coomber and Barriball [26] stated that nurses’ satisfaction and stability is more influenced by their work environment than by individual and demographic factors. This finding implies that attention should also be paid to the study of psychosocial factors in the work environment. According to Spence Laschinger et al. [27], FLNMs currently face increased work demands, due to a large margin of control, which has reduced their visibility and direct impact on the support and mentoring of nursing staff. This situation leads to frustration for both nurses and FLNMs and threatens the quality of working relationships. Given the importance of this relationships for nurses’ satisfaction and also the quality of patient care, every effort must be made to create satisfactory working conditions for FLNMs. Highly perceived organizational support has been identified as an important predictor of job satisfaction and empowerment of managers [28].

The boundary of knowledge needs to be extended by the effects of these variables, i.e., communication skills (CS) and transformational leadership style (TFL), on nurses’ job satisfaction (JS), and the mechanism by which the relationships of all identified variables work. The implications of many studies in the field of managerial skills of FLNMs point to the need for their development, due to their insufficiency [2,8].

In addition, professional discussions on medical practice are appearing more and more frequently. The subject of the discourse is Slovak legislation, which allows, after a modification in 2018, a substitution of specialized management study for managers after serving 15 years. Such an experience is therefore equivalent to a specialized management study. Due to the workload of FLNMs, there is a lack of interest in supplementing their managerial knowledge and skills. In Slovakia, nurses have just one semester course of management as part of their undergraduate studies, where they become acquainted with the basics in this area. A frequently discussed topic is also decentralization and efforts to streamline the healthcare system, which leads to increased management margins and negative consequences for both staff and patients.

These facts are the basis for the construction of our research model (Figure 1), which aims to examine the relationship between CS and TFL, FLNMs on the JS of nurses managed by them and to verify the influence of three moderators (management practice, span of control and psychosocial work factors (PWF)) on the strength of this relationship.

From the abovementioned, there are four fundamental objectives in this research: (1) to verify that the transformational leadership style (TFL) of FLNMs is positively associated with the job satisfaction (JS) of nurses; (2) to verify if the communication skills (CS) of FLNMs are positively associated with the job satisfaction (JS) of nurses; (3) to examine the effects of the number of years of management practice of FLNMs on the relations of CS and TFL with JS of nurses; (4) to examine the effects of the span of control on the relationships of CS and TFL with JS of nurses.

## 2. Overview of Research

### 2.1. Job Satisfaction as Performance Outcome Measure and Related Management Processes

The patient’s perception of the level of quality of healthcare can be influenced by the nurse’s job satisfaction. Job satisfaction is defined as a nurse’s attitude in response to work experience and their compliance with her values and expectations [29]. Several studies clearly confirmed that job dissatisfaction is adversely reflected in nurse attitudes and behaviors and affects nurse–patient interactions and patients’ perceptions of these interactions [19,30]. Understanding this relationship is essential to address current and future shortages of medical staff [31]. The presented studies [7,32] reported findings on the relationship between FLNMs leadership skills and nurses job satisfaction. Factors that influence nurses’ job satisfaction include adequate staffing [33], senior leadership style [34,35], degree of decision-making autonomy [7,23,36], approach to information, the extent of control and the level of nurse–physician cooperation [32]. Psychological empowerment and workplace empowerment are also frequently mentioned factors [37,38,39,40]. Previous studies have also shown that stress [41], leadership and pay, workload, lack of time and lack of recognition have been cited as the most common reasons for nurses to leave their jobs [24,42]. Job dissatisfaction occurs when FLNMs fail to properly appreciate and support nurses, ignore staffing issues and neglect to address them. According to Coomber and Barriball [26], an examination of the sources of job dissatisfaction over time has revealed that the most important factors for the stability of nurses are still the work environment and not individual or demographic factors. Many studies have examined and confirmed the relationship between nurses’ job satisfaction and the quality of healthcare perceived by patients [7,19,20,31], as well as nurses’ retention [26,38,41,43] and their burnout [24,44]. In accordance with Kalisch [13], we assume that higher job satisfaction of nurses will result in cost savings, because high JS is associated with lower turnover [25,45,46]. Therefore, we can reasonably think of the JS of nurses as an indicator of performance in the nursing environment and examine its relationship with related management processes.

#### 2.1.1. Leadership Style

Leadership is a set of skills and abilities that a leader embodies and uses to formulate a vision and engage others to share it [1,47]. Leadership in the healthcare environment can be considered in relation to the role that the leader plays [48]. Research points out that using only one style of leadership is not appropriate; several elements need to be combined to ensure care processes that contribute to patient satisfaction, such as clear standard of care and role expectations, as well as collaboration [49]. Boyatzis and McKee [50], Laschinger et al. [27], Gilmartin and D’Aunno [51] and Peterson et al. [52] called for the need to pay attention to other theories of leadership that may be relevant to nursing and healthcare and are associated with positive individual and organizational results, including work performance, e.g., authentic leadership, resonant leadership and leader–member exchange theory. Due to the status, hierarchical, fragmented and interdisciplinary structure of the healthcare system not only in Slovakia, but also in other countries, there is so-called clinical leadership and clinical nursing leadership. This includes four key areas: facilitating effective ongoing communication, strengthening relationships within and among departments, building and maintaining teams and supporting employee involvement, thus helping to optimize healthcare performance [53,54]. Wong and Cummings [49] discussed the formal clinical management of FLNMs; other authors [55,56,57] drew attention to their informal leadership, both of which have an impact on the quality of healthcare provided. Wong, Cummings and Ducharme [49] presented a systematic review of leadership styles in nursing. Their conclusions indicate a large number of researched leadership styles with a predominance of transformational leadership style (TFL), but with rather ambiguous results and insufficient articulation of mechanisms by which leaders influence the various outcomes examined. Transformational leadership has been identified as the key to high performance and organizational efficiency in Morsiani et al. [7], Feather et al. [11], Kodama et al. [58], Doran et al. [9], Larrabee et al. [31], Casida and Pinto-Zipp [59] amd McCutcheon [60]. Transformation leaders have a significant positive impact on employee satisfaction through the provision of support, encouragement, psychological empowerment [61], positive feedback, support for open communication [9,61] and showing respect [11]. This leadership behavior tends to create a favorable environment in the workplace, which is characterized by increased cooperation, teamwork and few interpersonal conflicts. These behaviors have also been found to reduce nurses’ feelings of stress [10] and emotional exhaustion and to increase nurses’ self-respect [62] and commitment [58]. The application of TFL at the level of FLNMs has been associated with an increase in the number of nurses, their stabilization, job satisfaction and empowerment [4,12].

#### 2.1.2. Communication Skills

Communication is an essential element of care at every level of nursing practice. Ineffective communication between healthcare team staffs can lead to damage of the quality of care [63]. It is, therefore, important that nurses’ managers create an environment that promotes good communication, and helps nurses to develop their communication skills formally and informally [15]. Continuous effective communication of FLNMs helps to develop relationships between nurses, relationships within and between health professions, in order to secure and seamlessly exchange information, engage employees and support teamwork. The study by Vasconcelos et al. [63] emphasized that the use of effective communication as a management tool is essential for sharing critical information and creating a work climate. This leads to more efficient care processes at the level of the health microsystem [53]. Bender [53] spoke of FLNMs as “consistent communication points“ in healthcare. They permanently obtain information from all elements of the medical microsystem and further distribute it, as needed, to those who need this information for their work. They should, therefore, have the appropriate communication tools and skills needed for effective information transfer. Sorbello [64] stated the authentic statement of FLNMs as follows: “I think the biggest thing I work on every day is communication. Trying to keep people all together on the same page is the biggest thing I do.” Based on the results of the Parsons and Stonestreet [16] study, effective communication, availability of FLNMs and their willingness to listen and provide support and their ability to form clear expectations and feedback were identified as important determinants of nurses’ job satisfaction. Laschinger [27] stated that functional communication supports the building of effective relationships, and is a source of informal power for FLNMs. According to Timmins [15], ineffective communication can jeopardize patient safety and the quality of healthcare provided. FLNMs should, therefore, have adequate communication skills; in particular, they must communicate openly, listen to nurses, pass on relevant information, involve nurses in decision-making and resolve conflicts when they arise [15]. Based on the above, we argue that both communication and leadership are essential antecedents of job satisfaction of nurses.

**Hypothesis** **1.**
*A transformational leadership style (TFL) of FLNMs is positively associated with the job satisfaction (JS) of nurses.*


**Hypothesis** **2.**
*The communication skills (CS) of FLNMs are positively associated with the job satisfaction (JS) of nurses.*


### 2.2. Moderating Processes: Span of Control, Practice in Leading a Nurse and Psychosocial Factors

The ability to lead effectively is also influenced by situational factors. In our study, we focused on examining the impact of three important potential moderators on the relationship of selected management skills of FLNMs and the JS of nurses, which are:Experience in managing FLNMs,Span of control,Psychosocial work factors of FLNMs.

#### 2.2.1. Management Practice

More years of experience as a head nurse in a management position should lead to more experience in the ability to communicate effectively and lead their subordinates. On the other hand, more experience and the associated older age or rigidity in the implementation of new approaches can lead to a reduction in job satisfaction of direct subordinates. The length of management practice appears as a variable in several studies. The relationship of nursing practice to job satisfaction was discussed by Shader et al. [43]. Vesterinen [8] examined its relationship with the applied management styles. In his study, he confirmed that FLNMs with more experience were rated as more effective in management.

**Hypothesis** **3.**
*The number of years of management practice of FLNMs will moderate the effects of both CS (Hypothesis 3a) and TFL (Hypothesis 3b) on JS of nurses; specifically, the effects will be stronger when the governing practice is high.*


#### 2.2.2. Span of Control

The span of control is defined as the number of persons directly managed by the manager. According to Meier and Bohte [65], there is a certain size at which the control range reaches its maximum capacity to be effective, and increasing the size above this capacity does not bring any value; it can even be harmful. Several studies [66,67,68] found that the extent of the results is affected by the extent of control. According to Gittell [69], groups with a wide control range (average control range 34) were significantly associated with lower performance compared to groups with a narrow control range (average control range of nine). McCutcheon et al. [60] reported that a larger control margin reduces the positive effects of transformational leadership styles on nurses’ job satisfaction and patient satisfaction. According to the findings of Holm-Petersen et al. [66], a high span of control leads to similar results as management-style deficiencies. Kalisch [68] also recommended considering strategies to reduce the size of nursing teams to increase their performance.

In our study, the span of manager control was evaluated as the total number of employees subordinated directly to the manager and was obtained from managers. The total number of employees (both full-time and part-time) was taken into account instead of fully-time equivalents, as these would not exactly take into account the number of people who were directly accountable to the manager. The span of control included all categories of staff directly subordinate to the manager.

**Hypothesis** **4.**
*The span of control will moderate the effects of both CS (Hypothesis 4a) and TFL (Hypothesis 4b) the JS of nurses; specifically, the effects will be stronger when the span of control is lower.*


#### 2.2.3. Psychosocial Work Factors of the Head Nurse

Several studies have identified the need for support infrastructure and its alignment with the organizational strategy to support clinical leadership [70,71,72], suggesting that leadership alone can only be as successful as the infrastructure that supports it [53]. Perceived organizational support stems from employees’ general belief that their organization values their contribution and care about their well-being [73]. According to Patrick and Laschinger [74], when organizations provide FLNMs with support program that give them control over the resources they need to run effectively and make decisions, most of these managers’ report that their work frustration is reduced; on the contrary, they feel the desire to innovate, take risks and use the appropriate resources to achieve their goals. If support is lacking, FLNMs are frustrated and unhappy with their roles. Therefore, the perception of organizational support can play an important role in retaining current FLNMs and potentially attract future leaders to managerial positions.

**Hypothesis** **5.**
*Psychosocial work factors (PWF) will moderate the effects of both CS (Hypothesis 5a) and TFL (Hypothesis 5b) the JS of nurses; specifically, the effects will be stronger when PWFs are higher.*


## 3. Materials and Methods

### 3.1. Data Collection and Sample

Our study represents a descriptive study, aiming to verify the four fundamental goals, mentioned in the introduction. It is based on comparing the managerial skills of FLNMs with JS of directly subordinate nurses and identifying factors which lead to higher JS. FLNMs are managers who represent superior nurses or head nurses with a direct influence on nurses within the organizational structure of hospitals in Slovakia.

All data were collected in the form of a questionnaire survey in five university hospitals in Slovakia in the period September–November 2020. The main survey was preceded by a three-year pilot survey on the main problems of nurses in relation to job satisfaction (conducted in 2017, 2018 and 2019 in teaching health management nurses). In addition to financial issues, which are widely discussed in Slovakia, several groups of problems have been identified. The communication skills of the direct superior and his leadership style were listed in the first and second place in the number of answers. For this reason, we have designed our model to include these variables. Nurse managers and their subordinates from different types of clinical areas were approached to participate in the study, and were explained the meaning and purpose of the study. A total of 132 responses were obtained from FLNMs with a mean age of 48.4 years (min. = 31, max. = 66, SD = 10.11) and an average management experience of 13.47 years (min. = 2 years, max. = 33 years, SD = 8.51). Of the 132 FLNMs, 12% had a secondary education, 37.7% had a baccalaureate degree and 50.3% had a master’s degree and 31.1% had completed specialization in management. At the same time, 1120 nurses were interviewed, who answered only one question concerning their job satisfaction. The research team then summarized the questionnaires of FLNMs and their subordinates, where the average of job satisfaction of nurses, belonging to the head nurse, was the data (output variable) about JS for the manager. Only those FLNMs questionnaires, in which were found out the average of JS of the group of nurses were relevant.

### 3.2. Measures

#### 3.2.1. Communication Skills (CS) and Transformational Leadership (TFL)

The evaluation of CS is based on a tool developed by Hopkinson et al. [75], which characterizes front-line nurse leader communication behaviors and was identified according to the principles of instrument development by DeVellis [76]. Unlike other constructs, it focuses on leadership communication in the healthcare setting and its impact on nurse empowerment. It contains eight items, with the internal consistency statistics measured by the authors being higher than 0.7 for each sub scale (Cronbach alpha range from 0.71 to 0.93). These items are: (1) comprehensibility (the ability to set clear goals, constantly share information, give clear instructions, explain, express ideas effectively), (2) listening (the ability to actively listen), (3) openness (the ability to communicate openly, transparently—what I think and say, both sides, share knowledge, communicate key questions, ask for nurses’ opinions), (4) feedback (ability to provide specific timely feedback and recognition), (5) empathy (ability to asking about feelings, discussing personal matters), (6) non-verbal communication, (7) paralanguage (ability to use positive paralanguage influences), (8) manner (ability to communicate by appropriate means). The CS variable is operationalized as a score, created based on the FLNMs responses to eight items, which are scaled using five-point Likert-type scales (1 = “never”, 2 = “seldom”, 3 = “some of the time”, 4 = “most of the time” and 5 = “always”). After reliability analysis, the Cronbach’s alpha of the C was 0.87 (eight items). The TFL item is operationalized as a score, created based on the responses of FLNMs in relation to the four dimensions of transformational leadership—intellectual stimulation, inspirational motivation, idealized impact and individual approach—which were measured using a 20-item scale. The Multifactor Leadership Questionnaire is considered the best validated measure of TFL [77]. Responses to individual items within the TFL characteristics were scaled on a five-point scale (1 = “very seldom” to 5 = “very frequently”). After the reliability analysis, the Cronbach’s alpha of the TFL was 0.94 (20 items).

#### 3.2.2. Psychosocial Work Factors (PWF)

The PWF item is operationalized as a score, given by FLNMs to 14 items under three indices: (1) job demand (five items, such as workload, work pace, decision latitude and competence), (2) job control (four items measuring to what extent an individual has influence, how the work will be carried out and decisions affecting their work) and (3) managerial support (five items measuring managerial support and appreciation and help in developing an individual’s professional competencies) [78,79]. The scale is well established with excellent psychometric properties (Cronbach’s alpha from 0.73 to 0.88). Responses to individual items within the PWF characteristics were scaled on a five-point scale (1 = “totally disagree” to 5 = “totally agree”). After the reliability analysis, the Cronbach’s alpha of the PWF was 0.96 (14 items). In addition to Cronbach’s alpha, we also determined the reliability using the Average Variance Extracted (AVE) and Composite Reliability (CR) [80]. Our measurement model is acceptable. All estimates are significant. They are all above 0.5; most are above 0.7. The AVEs for all constructs are above 0.5 and the CR for all constructs is above 0.7.

#### 3.2.3. Job Satisfaction (JS)

One global item was used to measure JS, containing the question: “To what extent are you currently satisfied with your work as a nurse?” Responses were scaled on a five-point scale from 1 = very dissatisfied to 5 = very satisfied. Global JS is very often used in surveys and in the healthcare environment [33,81,82]. Global JS was positively correlated with job satisfaction, examined through satisfaction with partial aspects of work and confirmatory factor analysis showed one major factor [83,84]. Control variables were age (in years), education (0 = secondary, 1 = university 1st degree, 2 = university 2nd degree) and completion of a specialization study in management (0 = no, 1 = yes), which were selected as control variables given their theoretical relevance and the possibility of their influence on the investigated relationships. Vesterinen [85] also worked with the control variables age, education, length of work experience as a nurse manager and updating education in his study on FLNMs. The control variables chosen by us were also used in previous studies. According to a study by Shader et al. [43], the age of nurses and the experience of nursing are related to their job satisfaction. Education is also, according to Kalisch et al. [86], a factor influencing nurses’ job satisfaction.

### 3.3. Data Analysis

All data were analyzed using the IBM SPSS 24.0 software package (University of Stanford, CA, United States). Cronbach’s Alpha coefficient, AVE and CR was used to assess the internal consistency of the scale’s reliability. Descriptive statistics and hierarchical regression analyses were performed to test the established hypotheses. We used Jeremy Dawson excel templates to construct moderation effects graphs [87]. The ANOVA variance analysis was used to analyze multiple dependencies. We have worked with a 5% significance level.

## 4. Results

Relationships between individual variables were determined using a correlation matrix, which also includes control variables (Table 1). Table 1 also provides basic descriptive statistics for the file.

Descriptive statistics indicate that the TFL variable received the highest rating (mean = 3.95, SD = 0.54). JS as an output variable was evaluated, with an average value of 3.20 and SD = 0.67. PWF received a higher average rating, but with a higher standard deviation (mean = 3.33, SD = 0.82) compared to CS (mean = 3.10, SD = 0.67). The lowest average rating was given to CS. It is clear from the correlation matrix that there are significantly significant positive correlations between all examined variables, which indicates the feasibility of further analyzes. JS communication skills are most influenced by FLNMs with a correlation coefficient of up to 0.965 at zero significance. TFL affects the JS equally significantly, with a correlation coefficient of 0.944. All three moderators demonstrate significance in a simple relationship with independent variables. These are the relationship between practice and CS/TFL, where the dependence is negative, which indicates higher values of CS and TFL at lower practice; the span of control and CS/TFL relationships, where the dependence is also negative, indicate higher CS and TFL values at lower span of control; PWF and CS/TFL relationships, where the dependence is positive, indicate higher CS and TFL values at higher PWF values. For the control variables age, education and specialization, significant correlations were found in relations with JS (−0.736), i.e., JS is higher for younger age of FLNMs. Age also negatively and significantly correlates with the variables CS, TFL and PWF. Moreover, education has significant positive correlations with the variables JS, CS, TFL and PWF, so higher education at the 2nd level reflects better management skills of FLNMs and the JS of nurses. Another control variable specialization in management significantly correlates with all examined variables, so the completion of specialization studies in management is highly beneficial for the practice of managing FLNMs. Hypotheses 1 (CS) and 2 (TFL) were tested by multiple regression analysis. The results are given in Table 2. The table is divided into three columns; in the first and second columns, each predictor is listed separately, and in the third column, the results of the regression analysis for both predictors are given simultaneously.

When separately monitoring the direct effect, each of the variables, i.e., CS and TFL, had a significant relationship on the JS of nurses (ß = 0.89 for CS, ß = 1.12 for TFL). Especially with TFL of FLNMs (H2) alone, the effect on JS values is high. Of the control variables, only the specialization in management was significant, with the direct effect of CS on the JS of nurses, while in TFL, the positive relationship was not significant. If both predictors entered the regression analysis at the same time, the effect of TFL decreased significantly (ß = 0.39), while the effect of CS decreased only minimally (ß = 0.67), indicating that the suppressive effect is due to a large correlation between predictors. Both hypotheses (H1 and H2) are supported. The other three hypotheses, concerning moderators of management practice, span of control and PWF, were tested using moderated multiple regression analysis. The results are presented in Table 3, in which each hypothesis is divided into two columns for the solution of part (a) and part (b).

The effects of both predictors (CS and TFL) on the JS of nurses are moderated by all three moderators. The nature of these effects is shown in the following figures, which gradually plot six interaction effects. The first moderator is the management practice of FLNMs. Head nurses in which higher CS values were found also received a higher job satisfaction rating from their subordinate nurses (β = 0.52). The given variable also includes the variable practice, in which the β coefficient (−0.03) is statistically significant. Its negative value indicates that more experience in managing FLNMs is associated with lower nurse JS. Figure 2 shows the moderating effect of management practice in the relationship between CS and JS, which is significant but low. A similar situation arises with the moderating effect of the practice in the management of FLNMs in the relationship between TFL and JS, where with a weaker direct effect (β = 0.21), the moderating effect (0.10) is weaker but significant. A lower number of years of practice slightly increases the effect of TFL on JS (Figure 3).

The second moderator was the variable span of control and its moderating effects in relation to both predictors and the JS of nurses, which are shown in Figure 4 and Figure 5.

The direct effects between both variables and the JS of nurses are strong and significant (β = 0.65). The span of control has a negative effect in both cases, i.e., higher CS and TFL values in FLNMs were found in a smaller number of subordinates. This moderator has a significant interaction effect on the direct relationship, but it is low (0.05 and 0.04, respectively). This means that the lower span of control very slightly strengthens the positive relationship between CS and JS, as well as between TFL and the JS of nurses. Due to the almost identical coefficients found, Figure 4 and Figure 5 are very similar.

The third moderator is PWF. The moderation effect is shown in Figure 6 and Figure 7.

We can see from the graphs that the moderation effects are higher than in the previous two moderators, although they are negative. The direct effects are high for both predictors (β = 0.77, β = 0.81); at the same time, the individual effects of the moderator on the JS of nurses are also higher—all are significant. In CS, the moderation effect is PWF (β = −0.30), which indicates that with higher PWF values, the positive effect between CS and the JS of nurses decreases. Additionally, with TFL, the moderation effect is negative (β = −0.25). It is slightly lower than in CS; the decline in the positive relationship between TFL and JS is slightly more modest.

## 5. Discussions and Conclusions

The aim of our paper is to examine the relationship between communication skills and the transformation style of FLNMs management with the job satisfaction of the nurses they manage and to verify the influence of three selected moderators on the strength of this relationship.

The direct effects of both predictors, CS and TFL, on nurses’ job satisfaction were identified as strong and significant. This finding is consistent with the findings of other studies [9,15,16,31,59,60,85]. The results of the research show that the job satisfaction of nurses is most significantly affected by the communication skills of their superiors, in which, however, the current level was also evaluated by the lowest score. FLNMs themselves evaluate higher leadership skills; they feel less confident in communication skills.

Only the direct relationships between CS and TFL and moderating variables, i.e., the length of practice of FLNMs, control margins and PWF, were also examined. Nurses with shorter experience evaluated their own communication and leadership skills better (the average length of experience of FLNMs involved in the research was 13.47 years). It can be assumed that younger FLNMs are more aware of the need for researched skills and their importance for practice, and therefore, also pay increased attention to their development. Older FLNMs are likely to rely more on their own experience and lag in developing their communication and leadership skills. The research results also confirmed the effect of span of control on CS and TFL in FLNMs, with lower levels appearing to be more advantageous. With lower numbers of subordinated nurses, FLNMs feel more confident and better at their role. In the case of PWF, their direct influence has also been demonstrated; if they are provided with support from senior management and have created suitable conditions for their work, FLNMs are better able to lead subordinate nurses and communicate with them.

In this research, our intention was also to identify the influence of moderating factors on the relationship between CS and TFL on JS managed nurses. The effect of the length of FLNMs management practice, span of control and PWF was identified as significant in all three cases, with different potency to direct effects. This means that the chosen control variables interfere with the influenced relationships.

The weakest force on the relationship between CS and JS is shown by the variable span of control. The lower management margin slightly reinforces the positive relationship between CS and JS. This finding corresponds to the results of the study by Doran et al. [9], which did not confirm the extent of control as a predictor of nurses’ job satisfaction, but this factor was identified as a moderator of the relationship between transformational leadership style and nurses’ job satisfaction, with the interaction explaining 79 percent deviation in the study. In contrast, McCutcheon et al. [60] stated that the span of control is directly related to satisfaction, and the greater the span of control, the lower the job satisfaction of nurses and patient satisfaction.

The length of management practice strengthens the positive effect of CS on JS significantly and slightly stronger than span of control. Shorter experience in managing FLNMs is associated with higher nurse JS. This finding is interesting because in some studies, higher management skills are associated with longer practice [85]. Our results point to the fact that the length of practice is not able to provide sufficient managerial skills of FLNMs, and thus, cannot replace managerial education. On the contrary, based on the results, it can be assumed that the practice itself rather fixes stereotypes in management without supplementing them with current knowledge, which negatively affects the job satisfaction of nurses. Therefore, we do not consider a legislative measure enabling the replacement of specialized managerial education of leading medical employees in Slovakia by a sufficiently long practice to be an optimal solution. Consistent with the opinion of McConell et al. [2], we are convinced that gaining professional managerial knowledge through the subsequent specialized study of medical management can help create a knowledge base of organizational procedures, which is a prerequisite for fulfilling the task of providing high quality and effective health care [2].

The moderating effect of PWF is significant and more pronounced than with previous moderators; however, it is negative. With higher PWF values, the positive effect between CS and the JS of nurses decreases. This finding is unexpected. Although psychosocial support is directly reflected in the improvement of communication and leadership levels on the part of FLNMs (this is how the FLNMs themselves assess it), the impact on increasing the satisfaction of managed nurses has not been demonstrated. Based on our study, we cannot sufficiently explain this proven effect, and its deeper examination could be the subject of subsequent research.

In the direct relationship between the TFL and the JS of nurses, the moderators had a similar effect. This means that lower span of control only slightly increases the positive effects of the relationship between TFL and the JS of nurses. The moderating effect of the lower management practice of FLNMs is slightly stronger. Shorter practice increases the positive effects of TFL on the JS of nurses. The highest moderation effect was found for PWF. The values turned out to be significant and negative. Lower PWFs significantly enhance the positive effects between TFL and the JS of nurses.

There are significant differences in the level of management between individual health care facilities; only relatively few of them achieve a high level of management and apply best-practice procedures. According to Mc Conell et al. [2], a higher level of management can be seen in university hospitals and hospitals, in which an MBA education is implemented. The subject of our research were the mentioned types of facilities. Our findings also open up a lot of room for increasing the quality and efficiency of healthcare provision through the introduction of effective management approaches.

### Implications

Our study has several practical implications for the practice of nurse management. The first important implication is the recognition that the communication skills and transformational leadership style of FLNMs are a significant and strong predictor of job satisfaction of nurses, even in a strictly structured, hierarchical and status environment of university hospitals. Within the development of communication skills, it is necessary to focus on listening, feedback and empathy, which are the most neglected and at the same time highly beneficial for the job satisfaction of nurses. Even other communication skills in the form of constant information sharing, effective expression of ideas, openness and transparency of communication or the use of its various methods cannot be underestimated, because they bring equally positive effects in the work environment. The results of our study showed that the Slovak FLNMs have the greatest shortcomings in the field of communication skills, which have the most significant impact on the satisfaction of nurses.

Leadership style is the tool that creates a background for the satisfaction of nurses at work. Among the attributes of the transformational style of leadership, the items of idealized influence and inspiring motivation were highlighted. Our study, therefore, has significant implications for recognizing that in an uncertain and dynamically changing environment, those aspects of management’s transformation style that lead to employee stability and anchorage and create unquestionable confidence in the manager come to the fore. Inspirational leadership inspires followers with emotional means, emphasizing trust and faith in the ability of employees, and stimulates enthusiasm for work among subordinates. For those FLNMs who are able to use different communication skills and a transformational leadership style at the same time, the job satisfaction of nurses increases significantly. This finding is also important in connection with the fact that especially the younger generation of nurses preferred leaders, who have skills in leading people [88].

The second important implication of our research is that experience in managing FLNMs is not enough to perform management activities. Managers with a few years of experience have higher positive effects in terms of their communication skills and leadership style on the job satisfaction of nurses. The reason for this finding is the completion of a specialized study in the field of management in these FLNMs. This fact brings to the Slovak healthcare system important knowledge about the need for a specialized management study for managers in healthcare, which was recognized as optional by the government in 2018 on the grounds that 15 years of professional experience is sufficient for a managerial role. Our findings point to the opposite. FLNM may be a top expert in nursing, but it will be very difficult to gain managerial skills through trial and error. We agree with the statement of Vesterinen et al. [85] that FLNMs need more theoretical training to meet expectations and develop their professional skills. In collaboration with universities, healthcare organizations should begin planning FLNMs training programs focused on strategic issues, leadership, job satisfaction and change management. Additionally, McCallin [89] recommended that preparation for FLNM tasks include postgraduate study and formal management education. Lifelong learning should become the standard for FLNMs, in line with new models of health leadership development [90].

The third implication of our research is hospital management. It is not appropriate to increase the management margin, because our findings have pointed to the fact that with increasing margin, the job satisfaction of nurses decreases. A lower management margin slightly reinforces the positive relationship between the communication skills and the transformational leadership style of FLNMs and the job satisfaction of nurses.

The last implication points to the area of psychosocial factors of work. Research has confirmed that these are directly related to the feelings of nurse JS, as well as positively affecting the leadership and communication skills of FLNMs. For this reason, it is essential for the management of medical facilities to focus on improving them. However, the research results did not confirm its influence as a moderating factor; on the contrary, if they are moderators in the relationship between CS and JS, as well as between TFL and JS, then higher PWF values mitigate this direct effect (i.e., they have a negative effect). This finding will require further investigation.

FLMNs, in today’s healthcare environment, must play a dual role: act in accordance with organizational strategies, while creating and maintaining a culture of trust and teamwork among employees, which helps organizations achieve their strategic goals [91]. At the same time, employees in these positions have significantly less experience and often do not have leadership skills; most FLMNs gain them through experience. Investments in the development of FLNMs’ leadership skills should become an integral part of the health care facility strategy. We agree with Morsiani et al. [7] and Leat and Porter [90] that the development of leadership is not just about improving an individual’s leadership skills, but is an essential part of the development of the organization as a whole. Progressive health systems, which invest in the development of leadership skills of the entire management, will have a more significant return on investment in terms of organizational efficiency.

However, the lack of FLNMs in the future is also posed as a serious problem. In our study, as well as in the study of Laschinger et al. [27], the average age of an FLNM is 48 years, which emphasizes the need to attract and prepare the next generation of these managers. It is, therefore, critical to provide them with development programs and a supportive work environment.

## 6. Future Research and Limitations

The presented study has several limitations. The first is the geographical limitation of the study to the territory of the Slovak Republic. However, the fact that the same problems of shortage and stabilization of nurses repeatedly arise at the international level suggests that the national context may be less important than is generally assumed. This means that findings unique in the Slovak context can be applied in other parts of the world. The second limitation is a relatively small sample of respondents (132) given the total number of FLNMs in Slovakia. On the other hand, the study included five large university hospitals in Slovakia, which could support the generalization of results for the Slovak state health care. The implementation of research only in the public sector is also a limitation. We are also aware of a possible limitation, resulting from the fact that in the modeled relationships, we focused only on examining the relationships between variables. To confirm the causality, it would be necessary to meet two conditions, namely accrual and exclusion of another option (we had this condition partially fulfilled by controlled effects, but not completely, as our data were not experimental, but rather stemmed from questionnaires, and formed a “convenience sample”). Therefore, we did not address these issues. In the future, our research can be moved to the level of causality research using dynamic panel regression, which will allow us to take into account the existence of endogeneity and more appropriately describe the ongoing process of adaptation over time, as in the case of a static panel. The fourth limitation is the FLNMs’ subjective view of their communication skills and leadership style, which could differ from reality, when interviewing their subordinate nurses. Consequently, future research may focus on nurses’ views on these areas of FLNM management skills. Finally, in addition to the factors concerned in this study, there may be other factors that may affect the examined relationships. In the future, other theories can be combined, and a comprehensive analysis can be performed from various perspectives.

## Figures and Tables

**Figure 1 healthcare-09-00346-f001:**
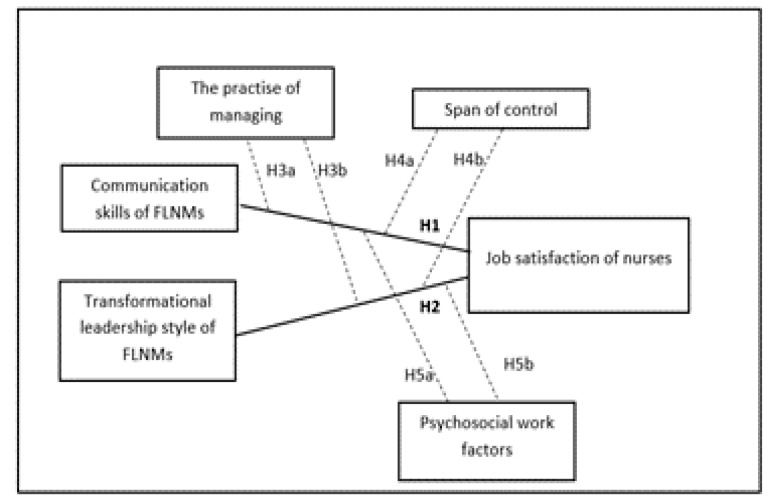
Model of the study.

**Figure 2 healthcare-09-00346-f002:**
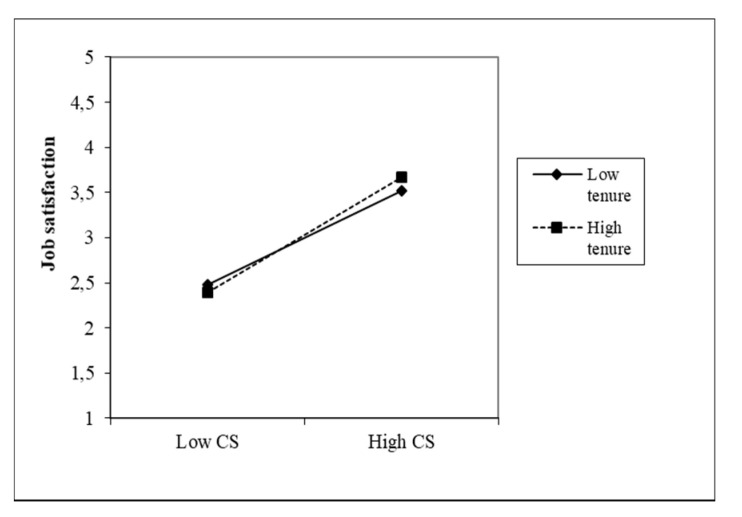
Moderating effect of management practice FLNMs on the CS–JS relationship.

**Figure 3 healthcare-09-00346-f003:**
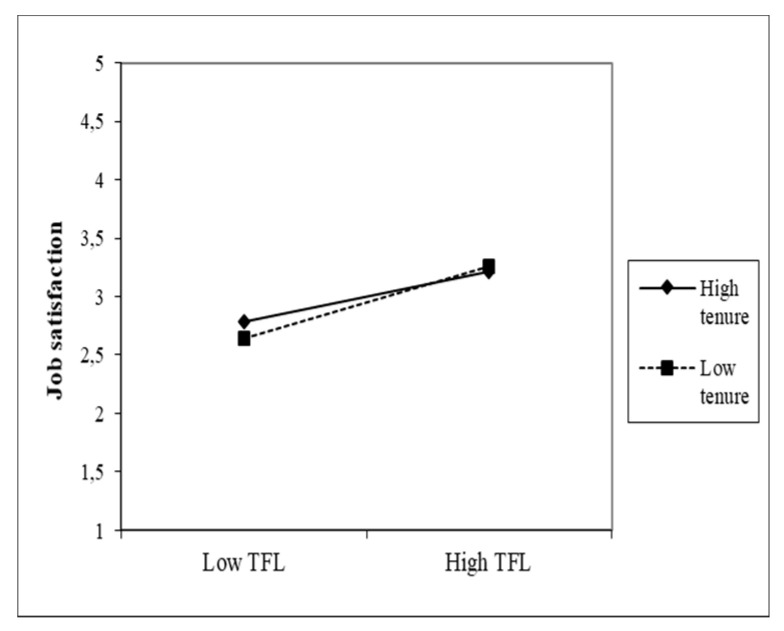
Moderating effect of practice in managing FLNMs on the TFL–JS relationship.

**Figure 4 healthcare-09-00346-f004:**
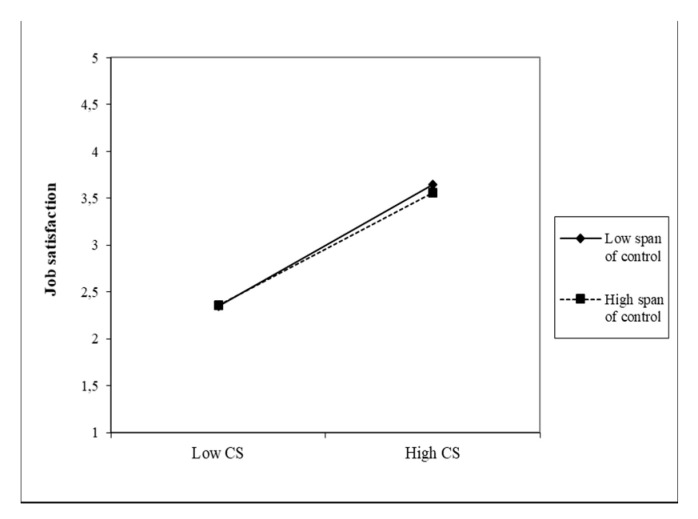
Moderating effect of the span of control on the CS–JS relationship.

**Figure 5 healthcare-09-00346-f005:**
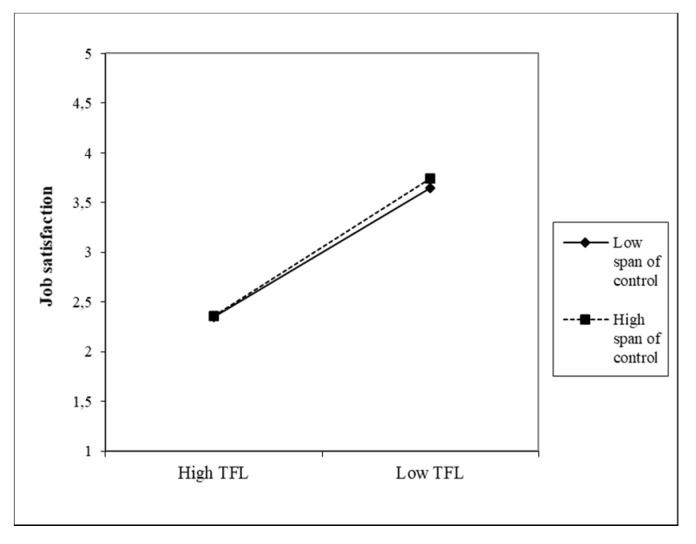
Moderating effect of the span of control on the TFL–JS relationship.

**Figure 6 healthcare-09-00346-f006:**
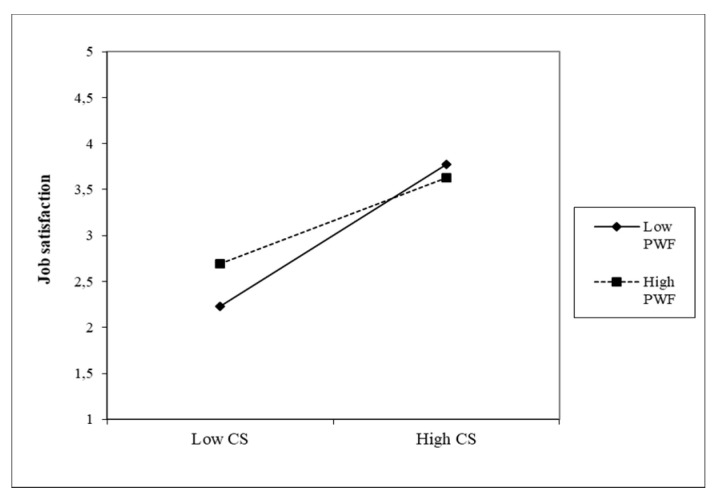
Moderating effect of PWF on the CS–JS relationship.

**Figure 7 healthcare-09-00346-f007:**
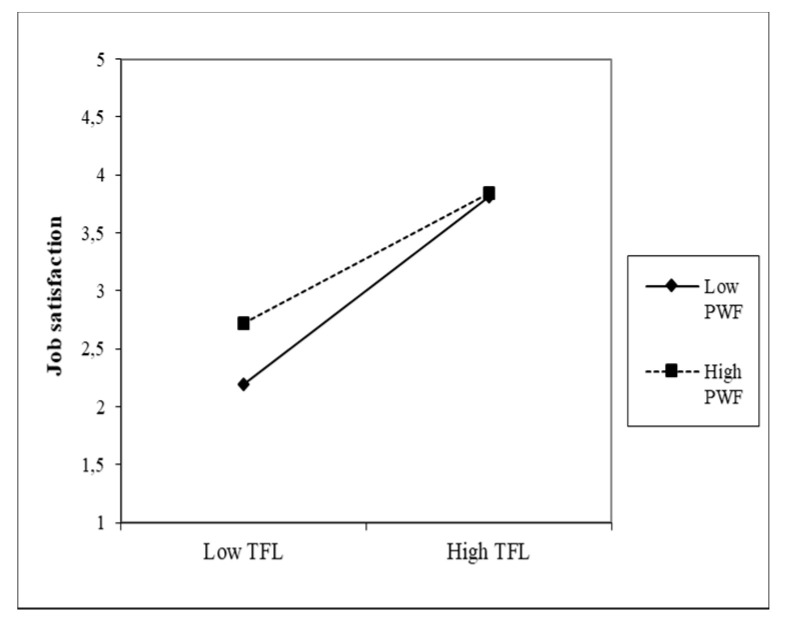
Moderating effect of PWF on the TFL-JS relationship.

**Table 1 healthcare-09-00346-t001:** Descriptive statistics of variables and correlation matrix.

Variable	N	Mean	SD	JS	CS	TFL	PWF	Age	Practice	Education	Specialization
JS	132	3.20	0.67								
CS	132	3.10	0.67	0.965 **							
TFL	132	3.95	0.54	0.944 **	0.898						
PWF	132	3.33	0.82	0.841 **	0.762 **	0.869 **					
Age	132	48.37	10.11	−0.458 **	−0.569 **	−0.326 **	−0.150				
Practice	132	13.47	8.51	−0.736 **	−0.817 **	−0.619 **	−0.435 **	0.832 **			
Education	132	1.18	0.69	0.741 **	0.667 **	0.803 **	0.735 **	−0.142 **	−0.291 **		
Specialization	132	0.31	0.46	0.558 **	0.434 **	0.684 **	0.765 **	0.297 **	0.096	0.804 **	
Span	132	27.31	4.44	−0.862 **	−0.830 **	−0.845 **	−0.824 **	0.478 **	0.740 **	−0.586 **	−0.432 **

Notes: JS = Job Satisfaction, CS = Communication Skills FLNMs, TFL = Transformational Leadership, PWF = Psychosocial Work Factors, Education = (secondary = 0, university 1st degree = 1, university 2nd degree = 2), Specialization in Management (no = 0, yes = 1), Span = Span of Control, ** *p* > 0.05.

**Table 2 healthcare-09-00346-t002:** Results of regression analyzes for two predictors (JS is the dependent variable).

	Hypothesis 1 (CS-JS)	Hypothesis 2 (TFL-JS)	Hypothesis 1 and 2 (CS+TFL-JS)
Constant	0.31	−0.7	−0.49
Main variables			
CS	0.89 **		0.67 **
TFL		1.12 **	0.39 **
Control variables			
Age	0.01	−0.01	0.00
Education	0.04	0.02	0.02
Specialization in management	0.2 **	0.02	0.06
R2 adj.	0.95	0.91	0.96

Notes: JS = Job Satisfaction, CS = Communication Skills FLNMs, TFL = Transformational Leadership, Education (secondary = 0, university 1st degree = 1, university 2nd degree = 2), Specialization in Management (no = 0, yes = 1), Span = Span of Control, ** *p* > 0.05.

**Table 3 healthcare-09-00346-t003:** Results of moderated regression analyzes.

	Moderator Practice		Moderator Span of Control		Moderator PWF	
	H3a	H3b	H4a	H4b	H5a	H5b
Control variables						
Age	0.01	0	0.01	−0.01	0	−0.01
Specialization in management	0.48	0.75	0.21	0.21	−0.01	0.22
Education		−0.06	0.01	−0.03	0.09	0.08
Main variables						
CS	0.52		0.65		0.77	
TFL		0.21		0.65		0.81
Practice	−0.03	−0.05				
Span of control			−0.04	−0.05		
PWF					0.16	0.28
CSxPractice	0.12					
CSxSpan			0.05			
CSxPWF					−0.30	
TFLxPractice		0.10				
TFLxSpan				0.04		
TFLxPWF						−0.25
Total R2	0.96	0.95	0.97	0.93	0.96	0.98

Notes: JS = Job Satisfaction, CS = Communication Skills FLNMs, TFL = Transformational leadership, PWF = Psychosocial Work Factors, Education (secondary = 0, university 1st degree = 1, university 2nd degree = 2), Specialization in Management (no = 0, yes = 1), Span = Span of control, *p* > 0.5.

## Data Availability

The data presented in this study are available on request from the corresponding author.

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
