# Peer review of "Communication Skills and Transformational Leadership Style of First-Line Nurse Managers in Relation to Job Satisfaction of Nurses and Moderators of This Relationship"

_healthcare, 2021, doi:10.3390/healthcare9030346_

Round 1

Reviewer 1 Report

Communication Skills and Transformational Leadership Style  of First-Line Nurses Managers in Relation to Job Satisfaction of Nurses and Moderators of this Relationship

 The introduction should be simplified and focus clearly on the themes it develops. It is necessary to develop in depth the leadership model assumed in this study. It is important to go deeper into this model in order to know its scope in the research.

Four hypotheses are proposed, which could become the four fundamental objectives of the research, since it is a descriptive study.

It is necessary to make explicit the design of the study and the specific target population (type of management positions, profile of the selected nurses, etc.).

Neither is it described how the sample (n=132) and (n=120) were calculated, nor the selection of the persons belonging to the sample. The inclusion and exclusion criteria are not adequately clarified.

It would also be very helpful to know the profile of the participants. You could include a table with the most significant variables.

As for the methods used through validated questionnaires, it is adequate. On the other hand, the discussion is well traced and reports all the results and their relation to practice.

I do not know the legislation of the country, but I am concerned that there is no reference to the study having been reviewed by a Research Ethics Committee. Likewise, it should be conveniently clarified whether this research meets the Helsinki criteria and whether each of the participants signed an Informed Consent prior to their participation in the research.

Author Response

Dear reviewer, thank you very much for the valuable comments, based on which we tried to incorporate to improve our contribution.

The introduction should be simplified and focus clearly on the themes it develops. It is necessary to develop in depth the leadership model assumed in this study. It is important to go deeper into this model in order to know its scope in the research.

We have partially shortened the introduction. The leadership model is described in more details, in the section "Overview of research" (2.1.1 Leadership style). In the introduction, we only outlined the current boundary of knowledge and importance of this topic in the field of health care management and special nursing, so that we could identify the research gap. We hope this will be okay.

Four hypotheses are proposed, which could become the four fundamental objectives of the research, since it is a descriptive study.

At the end of Chapter 1, we presented these 4 fundamental objectives. 

It is necessary to make explicit the design of the study and the specific target population (type of management positions, profile of the selected nurses, etc.).

In section 3.1 we added the concept of the study and the type of nurses managers. A sample is described below and it states, that the nurses come from different types of clinical areas. 

Neither is it described how the sample (n=132) and (n=120) were calculated, nor the selection of the persons belonging to the sample. The inclusion and exclusion criteria are not adequately clarified.

This remark was misunderstood because the research was conducted simultaneously between managers (n = 132) and their subordinate sisters (n = 1120) - of course anonymously. Each answer from the nurse about her job satisfaction was assigned to her senior nurse's questionnaire. Subsequently, the average job satisfaction was calculated for each manager, which was at the same time a certain feedback for her.  

It would also be very helpful to know the profile of the participants. You could include a table with the most significant variables.

 The profile of the respondents was examined with nurse managers and is described verbally in section 3.1 Data Collection and Sample. Of course, if the reviewer would require to prepare a table, we can do it. We didn't want to overcrowd the post with tables.

I do not know the legislation of the country, but I am concerned that there is no reference to the study having been reviewed by a Research Ethics Committee. Likewise, it should be conveniently clarified whether this research meets the Helsinki criteria and whether each of the participants signed an Informed Consent prior to their participation in the research.

We enclose the informed consent used in the data collection in the research.

The text of the article has passed the language check.

Reviewer 2 Report

The aim of our paper is to examine the connection between communication skills and the transformation style of FLNMs management with the job satisfaction of nurses and to verify the influence of 3 moderators on the strength of this relationship. It is an interesting paper. However, there are some suggestions for the author as follows.

  1. The introduction chapter, the sentence is too long to be easy to read, please divide into paragraphs.
  2. The literature cited in this article is too old, and it is appropriate to cite more than five years of journal articles. The author should review more recent relevant literature to improve it academic background.
  3. The author can draw a research structure diagram to help readers understand the research structure and hypothesis.
  4. The author can add a Participant Demographics table to illustrate the distribution of the sample.
  5. The article worked with a 5% significance level. The significance level list in table 1,2,3 should be *:p<0.05; **:p<0.01
  6. The descriptions in Figure 1 to Figure 4 are cut off, please correct.

Author Response

Dear reviewer, thank you very much for the valuable comments, based on which we tried to incorporate to improve our contribution.

1. The introduction chapter, the sentence is too long to be easy to read, please divide into paragraphs.

we divided into paragraphs

2. The literature cited in this article is too old, and it is appropriate to cite more than five years of journal articles. The author should review more recent relevant literature to improve it academic background.

older sources have been replaced and current sources have been supplemented

3. The author can draw a research structure diagram to help readers understand the research structure and hypothesis.

the research model has been added to the text

4. The author can add a Participant Demographics table to illustrate the distribution of the sample.

The research was carried out simultaneously between managers (n = 132) and their subordinate sisters (n = 1120) - of course anonymously. Each answer of the nurse about her job satisfaction was assigned to the questionnaire of her superior nurse. Subsequently, the average job satisfaction was calculated for each manager, which was at the same time a certain feedback for her. The profile of the respondents was examined with nurse managers and is described verbally in section 3.1 Data Collection and Sample. Of course, if the reviewer would require to preparing a table, we can do it. We didn't want to overcrowd the post with tables.

5. The article worked with. The significance level list in table 1,2,3 should be *:p<0.05; **:p<0.01

We only worked with a 5% significance level.

6. The descriptions in Figure 1 to Figure 4 are cut off, please correct.

Corrected

The text of the article has passed the language check.

Round 2

Reviewer 1 Report

I thank the authors for their efforts to improve the article. 
At this time I have only 2 concerns regarding its publication:
On the one hand the authors still do not provide the reference of the Ethics Committee that authorized the study and I believe that this is a universal criterion for publication in any high impact journal.
On the other hand, the structure of the article seems to me confusing with the reappearance of the 4 hypotheses. I think they should organize the results based on the objectives and discard the previous hypotheses from the text. 

Author Response

Thank you very much for accepting our previous edits. As for the reference of the Ethics Committee, we have not yet mentioned it in these types of articles, because it is not a clinical study, but a research of employees who confirmed their agreement to participate in the survey.

As for the 4 hypotheses, they are part of the research model itself, shown in Figure 1. At the same time, each of them is gradually developed in the theoretical part. If we removed them, it would significantly disrupt the structure of the article. If the reviewer and the editors insisted on removal, we would then remove the hypotheses.

Reviewer 2 Report

The article worked with a 5% significance level. The significance level list in table 1,2,3 should be *:p<0.05; **:p<0.01

Author Response

Thank you very much for accepting our previous edits.

As for the level of significance, we worked at the 5% level. The consultation of statistical methods for processing research results showed that it is not necessary to work with both levels, so we decided to do so.

Round 3

Reviewer 1 Report

The evaluations of the work have not changed from the previous review, so I think it will be up to the editor to decide if he wants to publish an article that does not have Ethics Committee approval. As for the hypotheses, I do not think it is adequate.

Reviewer 2 Report

Although the paper was not revised completely, but in its actual form can be considered for publication.